# Plasma Level of Pyrophosphate Is Low in Pseudoxanthoma Elasticum Owing to Mutations in the ABCC6 Gene, but It Does Not Correlate with ABCC6 Genotype

**DOI:** 10.3390/jcm12031047

**Published:** 2023-01-29

**Authors:** Eszter Kozák, Jonas W. Bartstra, Pim A. de Jong, Willem P. T. M. Mali, Krisztina Fülöp, Natália Tőkési, Viola Pomozi, Sara Risseeuw, Jeannette Ossewaarde-van Norel, Redmer van Leeuwen, András Váradi, Wilko Spiering

**Affiliations:** 1Institute of Enzymology, Research Center for Natural Sciences, Hungarian Academy of Sciences Center of Excellence, 1117 Budapest, Hungary; 2Department of Radiology, University Medical Center Utrecht, Utrecht University, 3584 CX Utrecht, The Netherlands; 3Department of Ophthalmology, University Medical Center Utrecht, Utrecht University, 3584 CX Utrecht, The Netherlands; 4Department of Vascular Medicine, University Medical Center Utrecht, Utrecht University, 3584 CX Utrecht, The Netherlands

**Keywords:** pseudoxanthoma elasticum, PXE, genotype-phenotype association, plasma pyrophosphate, ectopic mineralization

## Abstract

Background: Pseudoxanthoma elasticum (PXE), a monogenic disorder resulting in calcification affecting the skin, eyes and peripheral arteries, is caused by mutations in the ABCC6 gene, and is associated with low plasma inorganic pyrophosphate (PP_i_). It is unknown how ABCC6 genotype affects plasma PP_i_. Methods: We studied the association of ABCC6 genotype (192 patients with biallelic pathogenic ABCC6 mutations) and PP_i_ levels, and its association with the severity of arterial and ophthalmological phenotypes. ABCC6 variants were classified as truncating or non-truncating, and three groups of the 192 patients were formed: those with truncating mutations on both chromosomes (*n* = 121), those with two non-truncating mutations (*n* = 10), and a group who had one truncating and one non-truncating ABCC6 mutation (*n* = 61). The hypothesis formulated before this study was that there was a negative association between PP_i_ level and disease severity. Results: Our findings confirm low PP_i_ in PXE compared with healthy controls (0.53 ± 0.15 vs. 1.13 ± 0.29 µM, *p* < 0.01). The PP_i_ of patients correlated with increasing age (β: 0.05 µM, 95% CI: 0.03–0.06 per 10 years) and was higher in females (0.55 ± 0.17 vs. 0.51 ± 0.13 µM in males, *p* = 0.03). However, no association between PP_i_ and PXE phenotypes was found. When adjusted for age and sex, no association between PP_i_ and ABCC6 genotype was found. Conclusions: Our data suggest that the relationship between ABCC6 mutations and reduced plasma PP_i_ may not be as direct as previously thought. PP_i_ levels varied widely, even in patients with the same ABCC6 mutations, further suggesting a lack of direct correlation between them, even though the ABCC6 protein-mediated pathway is responsible for ~60% of this metabolite in the circulation. We discuss potential factors that may perturb the expected associations between ABCC6 genotype and PP_i_ and between PP_i_ and disease severity. Our findings support the argument that predictions of pathogenicity made on the basis of mutations (or on the structure of the mutated protein) could be misleading.

## 1. Introduction

Physiological calcification is regulated through a complex network of calcification promoters and inhibitors which are finely tuned to confine calcification to the bones [1] and prevent pathological soft tissue calcification. Inorganic pyrophosphate (PP_i_) is a strong inhibitor of ectopic calcification and the effect of its dysregulation is demonstrated in pseudoxanthoma elasticum (PXE, OMIM 264800) [2,3], a model disease of soft tissue calcification.

Due to mutations in the ABCC6 gene, PXE patients have markedly reduced levels of circulating PP_i_ and as a consequence, ectopic calcification develops in multiple organs [2]. In the skin, calcification results in plaques and papules known as pseudoxantomas [4]. Calcification of Bruch’s membrane underlies peau d‘orange and angioid streaks in the eye, which cause considerable visual morbidity due to choroidal neovascularizations (CNV) and macular atrophy [5]. Peripheral arterial calcifications may result in peripheral arterial disease, gastric bleeding and microvascular brain damage [6,7].

Several proteins are involved in maintaining local and systemic PP_i_ levels, and disruption in one of these proteins results in increased ectopic calcification [8]. In PXE, more than 300 disease-causing ABCC6 variants have been found, with different effects on the protein [9]. ABCC6 protein is mainly expressed on the basolateral membrane of hepatocytes and is thought to be the main source of systemic PP_i_. The ABCC6 protein is responsible for increasing extracellular ATP, which is converted to PP_i_ and AMP [2]. PP_i_ levels therefore seem to be the causal link between ABCC6 variants and the multi-organ calcification observed in PXE. However, currently there are no data available on how ABCC6 genotype affects plasma PP_i_ levels. The present study therefore aimed to compare plasma PP_i_ between a large cohort of PXE patients and controls, and to investigate arterial and ophthalmological phenotypes of PXE with special emphasis on any association between genotype and plasma PP_i_ levels of the PXE patients who had biallelic pathogenic ABCC6 mutations.

## 2. Materials and Methods

### 2.1. Participants

PXE patients were recruited from the Dutch Expertise Center for PXE in the University Medical Center Utrecht, the Netherlands. Controls were recruited from the acquaintances of the PXE patients, excluding first and second-degree relatives. All patients had a clinical diagnosis of PXE based on the criteria of Plomp et al. [4]. In short, patients had to have two of the following three criteria: skin involvement (pseudoxanthomas), eye involvement (peau d’orange and/or angioid streaks), or genetic confirmation (biallelic variants in the ABCC6 gene). For this study, blood was collected for PP_i_ analysis, and clinical and molecular data from regular care were used (genetics, CT scans, exercise ankle-brachial index, skin photography, ophthalmological examination and imaging). This cross-sectional study was conducted in adherence to the tenets of the Declaration of Helsinki and approved by the medical ethical review board of the University Medical Center Utrecht (IRB# 18-767, 16-622). All participants gave written informed consent for blood collection and all the PXE patients gave written informed consent for the use of their medical files for research purposes.

### 2.2. Genotyping

Genomic DNA was isolated from whole blood. All ABCC6 exons and the flanking intron sequences were analyzed as part of the genetic screening for regular clinical care. Sanger sequencing was performed to identify rare sequence variants and small deletions and insertions, and multiplex ligation-dependent probe amplification (MLPA) was performed to screen for larger deletions in the ABCC6 gene (reference sequence NM_001171.5, MLPA kit P092B (https://www.mrcholland.com/), accessed 14 January 2021). Specific primers were used to avoid the amplification of ABCC6 pseudogenes ψ1 and ψ2 [10]. The pathogenicity of all ABCC6 variants was estimated based on the Sherloc criteria [11], a refinement of the guidelines for variant classification of the American College of Medical Genetics and Genomics Association for Molecular Pathology, and classified as benign (class 1), likely benign (class 2), variant of unknown significance (VUS, class 3), likely pathogenic (class 4) or pathogenic (class 5), as previously described [12]. Patients with benign and likely benign variants were excluded from the analysis. VUS with a tendency towards benign or pathogenic, as based on the Sherloc criteria [11], were classified as such. Since all patients had a confirmed clinical diagnosis of PXE, based on [4], we included patients with a VUS in our analyses. Variants were classified as truncating (nonsense, out-of-frame insertions or deletions, splice site variants that result in a frameshift and deletions of the whole ABCC6 gene) and non-truncating (missense variants, in-frame insertions or deletions and splice site variants that result in the lack of an in-frame exon), and patients were classified as having two truncating variants, a mixed genotype of one truncating and one non-truncating variant, or two non-truncating variants.

### 2.3. Blood Collection and PP_i_ Analysis

Blood was collected in 4.5 mL CTAD vacutainers (BD, 0.11 M buffered trisodium citrate solution, 15 M theophylline, 3.7 M adenosine and 0.198 M dipyridamole). 50 µL of a 15% K3-ethylenediaminetetraacetic acid (EDTA) solution was added and the tubes were centrifuged for 15 min (1000× *g*, 4 °C). Blood plasma was transferred to a Centrisart ultrafiltration tube (300 kDa, Sartorius, Göttingen, Germany), centrifuged for 30 min (2200× *g*, 4 °C) and the filtered plasma was stored at −80 °C until PP_i_ analysis. Plasma PP_i_ concentrations were determined as previously described [2]. In short, PP_i_ was converted into ATP using ATP sulfurylase in the presence of excess adenosine 5′-phosphosulfate (APS). For each 10 μL of plasma, 70 μL of a mixture containing 32 mU ATP sulfurylase (ENZ-353; ProSpec, East Brunswick, NJ, USA), 16 μmol/L APS (A5508; Sigma-Aldrich, St. Louis, MO, USA), 80 μmol/L MgCl_2_, and 50 mmol/L HEPES (pH 7.4) was added. Following incubation at 37 °C for 30 min, ATP sulfurylase was inactivated at 90 °C for 10 min. Generated ATP was quantified using the ATP-monitoring reagent BacTiter-Glo (G8230; Promega, San Luis Obispo, CA, USA). An equal volume of BacTiter-Glo reagent was added to the samples. Bioluminescence was determined in a microplate reader (EnSpire Multimode Reader; PerkinElmer, Waltham, MA, USA). Each sample was measured twice in triplicates.

### 2.4. Arterial Phenotypes

#### 2.4.1. Arterial Calcification

Arterial calcification mass was measured on unenhanced low dose (< 3 mSv for a 70 kg adult), whole body CT scanners (Siemens or Philips, mAs dependent on body weight, 100 or 120 kVp, slice thickness < 1 mm which were resampled to 5 mm slices with 4 mm increment) in the carotid siphon, common carotid artery, coronary arteries, thoracic and abdominal aorta, iliac arteries, and femoral and crural arteries. Arterial calcification was defined as hyperdense arterial wall lesions with a density above 130 Hounsfield Units (HU). Calcification mass scores were computed as the product of the volume of the lesion in ml and the mean attenuation in HU. A peripheral artery score was calculated as the sum of the femoral and crural arteries, and a total arterial calcification mass score was calculated as the sum of all different arterial beds.

#### 2.4.2. Ankle-Brachial Index

The ankle-brachial index (ABI) after a treadmill test was used to measure PAD. To calculate the ABI, systolic blood pressure was measured in the left and right brachial arteries, the tibial posterior arteries and the dorsal pedal arteries. During the treadmill test, patients were encouraged to walk on a treadmill with a 10% slope and a speed of 3.5 km/h for 6 min. Patients who stopped prematurely due to claudication were encouraged to continue walking as soon as possible. During recovery, the ABI was measured for 10 min in a supine position. The lowest value of these ABI measurements was used as the exercise ABI. The ABI measurements were performed by experienced technicians. PAD was defined as an ABI < 0.90. Patient-reported claudication was measured with the Fontaine classification and patients were classified as having no PAD (ABI > 0.90), asymptomatic PAD (ABI < 0.90 without claudication) and symptomatic PAD (ABI < 0.90 with claudication).

### 2.5. Ophthalmological Phenotypes

All patients underwent routine ophthalmological examination, including best-corrected visual acuity (BCVA), macular spectral domain optical coherence tomography (SD-OCT), fundus autofluorescence (FAF), near-infrared reflectance imaging (NIR) (all Spectralis HRA-OCT, Heidelberg Engineering, Heidelberg, Germany) and color fundus photography (FF 450 plus, Carl Zeiss Meditec AG, Jena, Germany).

#### 2.5.1. Length of Angioid Streaks

The length of angioid streaks on 55° near infrared (NIR) imaging was used as a proxy for the extent of Bruch’s membrane calcification, as previously described [5]. The length of the longest angioid streak from the center of the optic disc to the retinal periphery was measured and graded into four zones: zone 1 (<3 mm from the center of the optic disc), zone 2 (3–6 mm), zone 3 (6–9 mm) and zone 4 (>9 mm).

#### 2.5.2. Macular Phenotype

The presence of CNV and the severity of macular atrophy were scored in both eyes by three experienced graders (SR, RvL, JOvN). The presence of (in)active CNV was based on assessment of SD-OCT, fundus photography and fluorescein or indocyanine green angiography. The presence of macular atrophy was based on FAF, SD-OCT, color fundus photography and NIR. The largest atrophic area was graded as ‘no atrophy’, ‘mild atrophy’ (<twice the area of the optic disc) or ‘severe atrophy’ (> twice the area of the optic disc). An atrophic zone surrounding an (in)active CNV was considered independent of macular atrophy. BCVA was measured on Snellen charts or Early Treatment of Diabetic Retinopathy Study (ETDRS) charts and converted to the logarithm of the minimum angle of resolution (logMAR).

### 2.6. In Vivo Expression of ABCC6 Protein in Mouse Hepatocytes

Animal protocols were approved by the Food Chain Safety and Animal Health Directorate of the Government Office of Pest County, Hungary (XIV-I-001/707-4/2012). ABCC6-expressing liver samples were obtained from 12-week-old *Abcc6*^−/−^ female mice by hydrodynamic tail vein injection (HTVI) of ABCC6 in pLIVE plasmid (Mirus Bio, Madison, WI, USA) as described [9]. Briefly, wild-type and mutant ABCC6 variants’ cDNA were cloned into pLIVE expression plasmid (Mirus Bio, Madison, WI, USA), and each animal was given 70 μg of the construct dissolved in 2 mL of sterile saline, injected in the lateral tail vein in no more than 5 s, achieving the necessary hydrodynamic shock that creates a temporary disruption in the membrane of hepatocytes. Four days later, the mice were euthanized by an overdose of anesthetic (intraperitoneal injection of tiletamine/zolazepam (30 mg/kg, Zoletil, Virbac, France), xylazine (12.5 mg/kg, Primazin, Alfasan, The Netherlands) and butorphanol (3 mg/kg, Butomidor, Richter Pharma, Austria), liver lobes harvested and snap-frozen in 2-methylbutane cooled by liquid nitrogen. Immunohistochemistry was performed as previously described [9].

### 2.7. Statistical Analysis

Characteristics are presented as mean ± standard deviation for normally distributed continuous variables, median (interquartile range (IQR)) for non-parametric distributed continuous variables and number (%) for categorical variables. For the ophthalmological phenotypes, data for the right eye were used for descriptive and regression analyses. If imaging of the right eye was not assessable, the left eye was used. The correlation between plasma PP_i_ and continuous outcome measures was tested with the Pearson correlation.

Linear regression models were built with genotype or the individual phenotypes as the determinant and plasma PP_i_ as the outcome, and adjusted for age and sex. Log_10_ transformation was performed on the calcification mass scores. The models including arterial calcification mass scores were additionally adjusted for differences in scanner and settings since this has been shown to affect calcification mass scores [13,14]. A *p*-value < 0.05 was regarded as statistically significant. All analyses were performed in RStudio version 1.1.456.

## 3. Results

In total, 207 PXE patients from 175 families and 45 controls were included in this study. Age was similar between the groups (PXE: 55 ± 13 vs. controls 52 ± 16 years, *p* = 0.23), but more of the PXE patients were females (125 (60%) vs. 82 (40%) males, *p* = 0.02). In all 207 patients and 45 controls, PP_i_ was measured. In addition, 189 patients underwent CT scanning, all patients underwent an exercise ABI measurement and 193 patients underwent an ophthalmological examination. Of the 207 patients, 192 were found to have biallelic pathogenic ABCC6 mutations.

### 3.1. Determinants of PP_i_ in PXE

The PXE patients had lower plasma PP_i_ than the controls (0.53 ± 0.15 vs 1.13 ± 0.29 µM, *p* < 0.01, Table 1). In the PXE patients, plasma PP_i_ was correlated with age (β: 0.05 µM, 95% CI: 0.03–0.06 per 10 years, r = 0.40, *p* < 0.01). The female PXE patients had slightly but significantly higher PP_i_ than the males (0.51 ± 0.13 in males vs. 0.55 ± 0.17 µM in females, *p* = 0.03, Figure 1). The difference between males and females remained statistically significant after adjustment for age (β: 0.06 µM, 95% CI: 0.02–0.10).

### 3.2. Association of PP_i_ and Arterial Calcification and Function

Higher peripheral and total arterial calcification mass were weakly correlated with higher plasma PP_i_ (r = 0.19, *p* < 0.01, and r = 0.25, *p* < 0.01, respectively). However, when adjusted for age and sex, no association between plasma PP_i_ and peripheral (β: 0.00 µM, 95% CI: −0.02–0.02) and total (β: 0.01, 95% CI: −0.02–0.04 µM) arterial calcification mass was found. Patients with PAD (*n* = 108, 52%) had higher plasma PP_i_ than patients without PAD (0.56 ± 0.16 with PAD vs 0.51 ± 0.15 µM without PAD, *p* = 0.01), but this difference attenuated after adjustment for age and sex (β: 0.02 µM, 95% CI: −0.02–0.06, Table 2).

### 3.3. Association of PP_i_ and Ophthalmological Phenotypes

Patients with CNV (*n* = 119, 62%), had higher plasma PP_i_ than patients without CNV (0.57 ± 0.15 with CNV vs. 0.48 ± 0.14 µM in patients without CNV, *p* < 0.01) and worse visual acuity was weakly correlated with higher PP_i_ (r = 0.19, *p* < 0.01, Figure 2). When adjusted for age and sex, the presence of CNV was associated with higher plasma PP_i_ (β: 0.06 µM, 95% CI: 0.01–0.11, Table 2). No significant association between PP_i_ and the other ophthalmological phenotypes was found.

### 3.4. Association of PP_i_ and ABCC6 Genotype

Of the 207 patients, 192 were included in the ABCC6 genotype and plasma PP_i_ level correlation analysis (for a detailed list, see Appendix A). ABCC6 variants were classified as truncating or non-truncating. Accordingly, three groups of patients were formed: those with truncating mutations on both chromosomes (*n* = 121), those with two non-truncating mutations (*n* = 10), and a third group who had one truncating and one non-truncating ABCC6 mutation (*n* = 61).

In our PXE cohort, a total of 81 non-truncating missense alleles were identified: 61 of these were in the ‘mixed’ group (i.e., the other allele had a truncating mutation), and the remaining 20 were the alleles of 10 patients carrying missense mutations on both chromosomes. Altogether we found 34 different missense mutations, which represented a spectrum with high variety.

No significant difference was found between the plasma PP_i_ levels of the groups with two truncating ABCC6 alleles, or two non-truncating alleles, or those who had one truncating and one non-truncating allele (Figure 3A).

We analyzed a few mutations in detail: for instance, the mutation R1141* was present in homozygous form in 21 (10.9%) of the 192 PXE patients, making R1141*/R1141* the most common ABCC6 genotype in the cohort examined. This high prevalence is not unexpected, since this mutation is quite common in the European PXE population [15,16,17]. It results in a C-terminally truncated ABCC6 lacking a large part of the protein, rendering it nonfunctional, and decreasing its expression to undetectable levels [18]. Figure 3B highlights patients homozygous for R1141* on the age vs. plasma PP_i_ plot, revealing that plasma PP_i_ concentrations observed in these patients varied widely (between 0.35 and 0.85 μM) and that the ABCC6 genotype of these patients apparently did not account for these differences. Even the three patients from the same family (shown as squares) had different plasma PP_i_.

We studied the cellular localization of two missense mutants by overexpressing the human ABCC6 protein variants in the liver of *Abcc6^−/−^* mice [9]. A strength of this experimental approach is that the expression happens in vivo in the native environment of the liver, in the tissue and cell type where ABCC6 protein is physiologically present. In addition to confirming protein expression, this method can also establish the subcellular localization of a given mutant in its physiological conditions. We have shown earlier that wild-type human ABCC6 protein is expressed in the basolateral plasma membrane of the hepatocytes of the mouse liver (see Figure 3E, central panel; this is a criterion of being physiologically active), while some mutants show complete or partial intracellular retention [9,19]. Missense mutant R1138Q was found in five patients of the mixed group, i.e., each patient carried this mutation on one allele and a truncated one on the other allele. The cellular localization of this mutant showed partial intracellular retention, as shown on Figure 3E, left panel, in harmony with our earlier observation [9]. The plasma PP_i_ level varied in these five patients between 0.40 μM to 0.96 μM, lacking any apparent correlation between this specific missense mutation and the plasma PP_i_ measured.

The mutation R1314Q, which also gave rise to a partially mislocalized ABCC6 protein, was found in a PXE patient with 0.35 ± 0.03 μM plasma PP_i_ concentration, and one who, in contrast, had 0.7 ± 0.04 μM of plasma PP_i_ (Figure 3E, right panel). The difference is striking, even though these patients’ ABCC6 genotypes are similar, with R1314Q on one allele, and a truncating mutation on the other.

## 4. Discussion

In this large cohort of PXE patients, we show that PP_i_ levels are approximately 60% lower than those of a control population, which confirms that PP_i_ is involved in the pathogenesis of PXE. PP_i_ levels were positively correlated with increasing age and were slightly higher in females. Similar results have been published recently, independent of the present study, which support these correlations being solid, the results coming from two different cohorts of 207 patients (present study) and 107 patients [20].

More importantly, we found that plasma PP_i_ levels were not associated with disease severity in PXE. Only a small association between plasma PP_i_ and CNV was found, and other ophthalmological outcomes were not associated with plasma PP_i_. In the recent publication of Leftheriotis et al. also no association between plasma PP_i_ and lower limb arterial calcification was found [20], comparable to our results concerning PP_i_ and peripheral artery calcification score. Total body arterial calcification, as well as ophthalmological outcomes, were not studied by these authors. Contrary to PP_i_, age seems to be the most important determinant for disease severity in PXE. More ectopic calcification, as seen in older patients, leads to more need for inhibitors of it, of which PP_i_ is one of the most important, but as we did not find an association between PP_i_ and disease severity it remains difficult to explain the association between PP_i_ and age that we and others have found.

Our findings add to a growing body of evidence showing that both systemic and local factors contribute to the pathophysiology of PXE. The ABCC6 protein is thought to account for approximately 60% of circulating extracellular PP_i_ [2]. Although studies of PXE patients and mouse models consistently show reduced systemic PP_i_ [2,21,22], PXE is characterized by large heterogeneity of symptoms in patients with the same pathogenic variants and even within families [23]. We recently showed that patients with two truncating variants in the ABCC6 gene have more severe arterial calcification and longer angioid streaks than patients with a mixed genotype, which emphasizes the role of ABCC6 in ectopic calcification. This, however, did not translate to more severe clinical consequences [12].

An interesting study has been published recently regarding the genotype, disease phenotype and plasma PP_i_ of patients with inborn calcification disorders caused by inactivating mutations of ectonucleotide pyrophosphatase/phosphodiesterase 1 (ENPP1) encoding the nuclease acting upstream to ABCC6 protein [24] (see Figure 4A). Mutations in ENPP1 are responsible for the severe disorder generalized arterial calcification in infancy (GACI). In this study, two individuals diagnosed with PXE were found to have biallelic mutations in their ENPP1 gene, yet surprisingly these patients presented a more favorable clinical outcome of PXE phenotype. Their plasma PP_i_ concentrations were substantially decreased, similar to the two individuals who had biallelic mutations in their ENPP1 genes and the much more severe phenotype of GACI. This observation also suggests that the relationship between reduced plasma PP_i_ and ectopic calcification symptom severity may not be as direct as previously thought.

In the present study we have found that even though biallelic ABCC6 mutations do decrease plasma PP_i_ level, the highly variable plasma PP_i_ concentrations measured in our PXE cohort cannot be attributed to the specific type of ABCC6 mutations these individuals harbor, even though ABCC6 protein has a key role in the metabolic pathway responsible for approximately 60% of circulating extracellular PP_i_ [2] (see Figure 4A). One of the strengths of the present study is the large number of patients included, with both detailed genotypic characterization and plasma PP_i_ data available.

The lack of association between genotype and plasma PP_i_ level in PXE revealed by the present study is rather unexpected, since one would have predicted that several missense ABCC6 variants preserved decreased residual activity, thus providing higher plasma PP_i_ than found in the truncating/truncating group. However, we detected no difference between the PP_i_ levels of patients with one or with two missense mutations as compared with the values of the truncating/truncating group. Furthermore, our finding that the same mutation is accompanied by very different plasma PP_i_ (see Figure 3) also mitigates against the above assumption. Our findings collectively suggest the existence of additional factors perturbing the expected associations between genotype and plasma PP_i_, and between plasma PP_i_ and disease severity. Our hypothesis on the role of potential factors is summarized in Figure 4. ABCC6 protein in the liver is thought to be the main source of extracellular ATP, which is subsequently converted to PP_i_ and AMP by ENPP1 [2]. Factors potentially perturbing the genotype–plasma PP_i_ correlation are: (i) the level of ABCC6 protein expression in the liver, controlled by e.g., oxidative stress [25,26]; and (ii) the activity of additional proteins involved in generation (ANK [27] and ENPP1) or removal (NT5E and TNAP) of extracellular PP_i_, which may significantly influence plasma PP_i_. In addition, there is competition for ATP in the circulation: ENPP1 is the only nuclease producing PP_i_ from ATP, while ectonucleoside triphosphate diphosphohydrolase (eNTPD) enzymes ‘consume’ ATP without generating PP_i_ [28]. This is another perturbing effect reducing the ATP substrate concentration available for ENPP1.

Furthermore, PP_i_ is not the sole possible inhibitor of ectopic calcification, and disease severity in PXE may be influenced by other calcification inhibitors, e.g., fetuin A [29]. We also have to consider the roles of environmental factors like diet (some food contains PP_i_ as an additive), or smoking, which could enhance disease severity.

Utilizing animal models, it was found that although reduced plasma PP_i_ is a major cause of ectopic mineralization in PXE, it is not the only factor that prevents ectopic calcification [30,31]. In spite of this complex picture, several therapies to increase PP_i_ are currently under development for the treatment of PXE and related disorders. Orally delivered PP_i_ has been shown to increase plasma PP_i_ in humans and mice, and to inhibit cardiac calcification and calcification of the vibrissae, kidneys and arteries of the legs in mice [22]. These latter results further support the causal role of reduced PP_i_ in PXE. The recombinant ENPP1 enzyme INZ-701 was shown to increase plasma PP_i_, inhibit cardiac and aortic calcification and increase survival in *Enpp1*^−/−^ mice [32], and in addition it also increased plasma PP_i_ and partially prevented calcification in *Abcc6^−/−^* mice [33]. Inhibition of TNAP also reduced calcification in *Abcc6*^−/−^ mice [34]. Finally, we have demonstrated that the bisphosphonate etidronate, a stable analogue of PP_i_, inhibits systemic arterial calcification in PXE patients [35].

## 5. Conclusions

Our results support that PXE patients have approximately 60% reduced plasma PP_i_ levels when compared with controls. Plasma PP_i_ levels were correlated with increasing age and were higher in female PXE patients. Contrary to expectations, no clear association between plasma PP_i_ levels and disease severity was found in PXE. Another unexpected finding of our study is that no correlation between ABCC6 genotype and plasma PP_i_ level was found in the 192 pseudoxanthoma elasticum patients with known pathogenic phenotypes. Our findings should trigger future studies of the role of PP_i_ in disease progression, and possible genetic or environmental modifiers, which might shed more light on the complex pathophysiology of PXE.

## Figures and Tables

**Figure 1 jcm-12-01047-f001:**
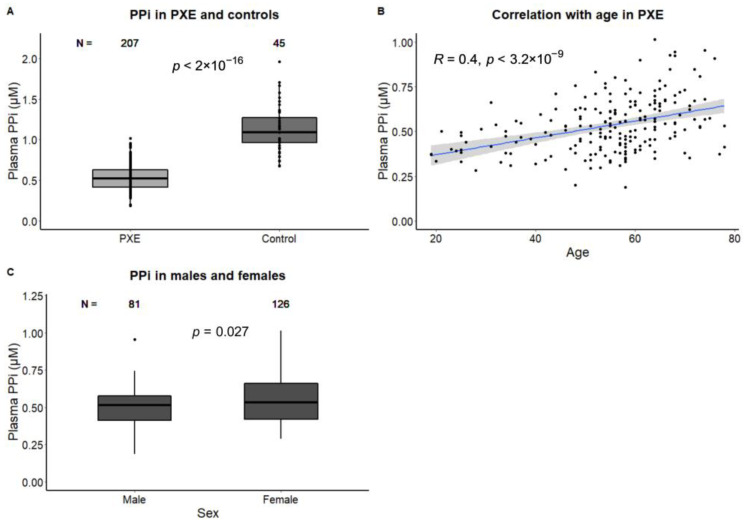
Inorganic pyrophosphate in PXE and controls. Plasma PP_i_ is approximately 60% lower in PXE patients than in controls (**A**). Plasma PP_i_ is correlated with age in PXE (**B**) and female PXE patients have slightly higher PP_i_ (**C**). Data were analyzed with Student’s *t*-test (categorical variables) or the Pearson correlation (continuous variables). A *p*-value < 0.05 was regarded as statistically significant.

**Figure 2 jcm-12-01047-f002:**
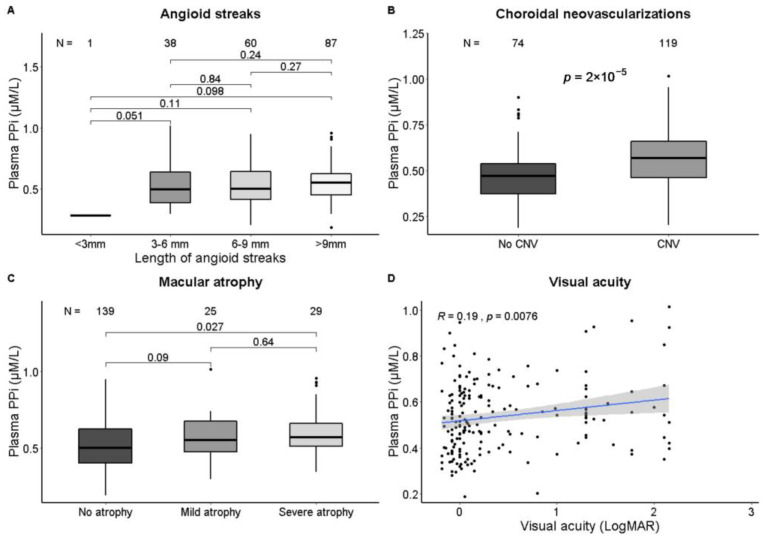
Correlation of plasma PP_i_ and ophthalmological phenotypes. Plasma PP_i_ is correlated with the presence of choroidal neovascularizations (**B**). Patients with severe macular atrophy have higher PP_i_ than patients without macular atrophy (**C**), and worse visual acuity is weakly correlated with higher PP_i_ (**D**). No significant correlation with the length of angioid streaks (**A**) was found. Data were analyzed with Student’s *t*-test (categorical variables) or the Pearson correlation (continuous variables). A *p*-value < 0.05 was regarded as statistically significant.

**Figure 3 jcm-12-01047-f003:**
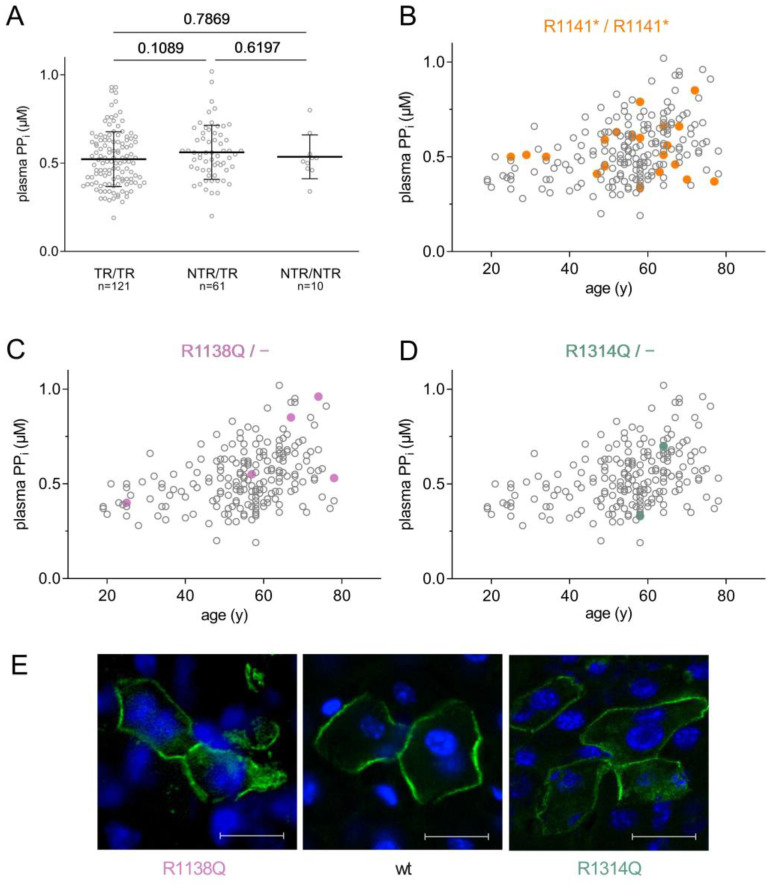
Correlation ABCC6 genotype and plasma PP_i_. (**A**) Plasma PP_i_ does not correlate with ABCC6 genotype. ABCC6 variants were classified as truncating or non-truncating. Accordingly, three groups of patients were formed: those with truncating mutations on both chromosomes (TR/TR); those with two non-truncating mutations (NTR/NTR), and a third group who had one truncating and one non-truncating ABCC6 mutation (TR/NTR). (**B**) Plasma PP_i_ levels of patients homozygous for R1141* vary between 0.35 and 0.85 μM. R1141* homozygotes are shown on the age vs. plasma PP_i_ plot as orange dots, R1141* homozygous patients from the same family are indicated by squares. (**C**) Patients with the non-truncating R1138Q mutation on one allele and TR on the other allele are highlighted in purple on the age vs. plasma PP_i_ plot; (**D**) patients with the non-truncating R1314Q mutation on one allele and TR on the other allele are highlighted in green. (**E**) Representative images of the subcellular localization of R1138Q, wt and R1314Q ABCC6 protein variants in vivo in the liver of *Abcc6*^−/−^ mice [10]; ABCC6 protein is shown in green, and nuclei stained by DAPI are shown in blue (scale bar = 20 µm).

**Figure 4 jcm-12-01047-f004:**
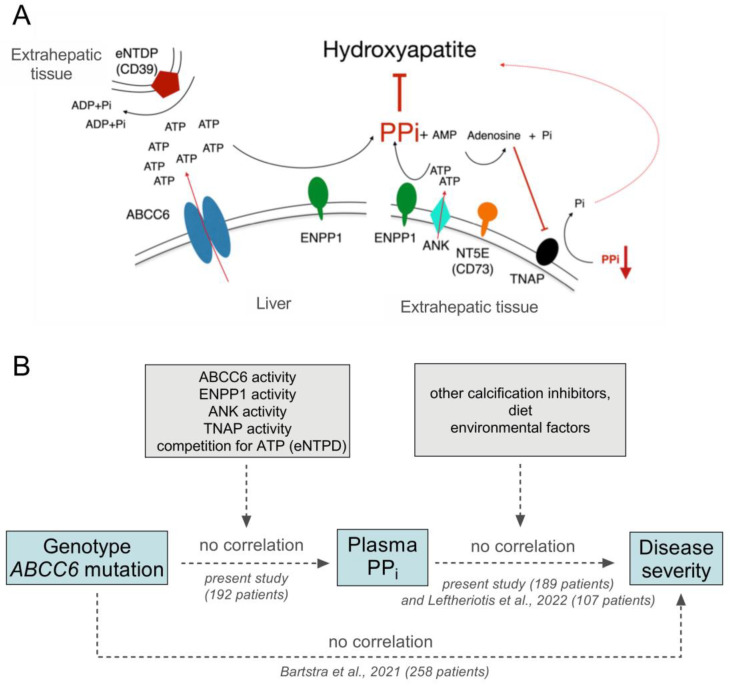
Crucial players of plasma pyrophosphate homeostasis, and factors modifying plasma pyrophosphate level and disease severity in pseudoxanthoma elasticum. (**A**) The activity of ABCC6 protein and ENPP1 in the liver are responsible for approximately 60% of extracellular PP_i_ in the circulation. In extrahepatic tissues, human progressive ankylosis (ANKH) protein, similarly to ABCC6 protein, facilitates ATP release from cells, while tissue-nonspecific alkaline phosphatase (TNAP) cleaves PP_i_, thus lowering its concentration. AMP is hydrolyzed by NT5E (CD73) to inorganic phosphate and adenosine, and the latter down-regulates TNAP expression, thus controlling its activity and increasing plasma PP_i_. Ectonucleoside triphosphate diphosphohydrolase enzymes (eNTPDs) compete for ATP with ENPP1. (**B**) Left: the expression and activity of ABCC6 protein, ANK and ENPP1 all contribute to increasing plasma PP_i_, while TNAP, on the other hand, decreases it. Competition for extracellular ATP in plasma by ENPP1 and eNTPDs may also influence pyrophosphate homeostasis. (**B**) Right: other calcification inhibitors, which exert their effect independent of plasma PP_i_ concentration, as well as dietary and environmental factors, can also impact calcification symptoms of PXE.

**Table 1 jcm-12-01047-t001:** Patient characteristics.

Characteristic	PXE (*n* = 207)	Controls (*n* = 45)	*p*-Value
Age, years	55 ± 13	53 ± 16	0.21
Sex, female	126 (61%)	19 (42%)	0.02
PP_i_, µM	0.53 ± 0.15	1.13 ± 0.29	<0.01
**Genotype**	***n* = 192**		
Two truncating, n (%)	116 (56%)		
Mixed, n (%)	71 (34%)		
Two non-truncating, n (%)	14 (7%)		
**Arterial calcification**	***n* = 189**		
Peripheral, mass score	589 (82–1932)		
Total body, mass score	1518 (186–4178)		
**Arterial functioning**	***n* = 207**		
Ankle brachial index	0.89 (0.63–1.04)		
Peripheral arterial disease, n (%)	108 (52%)		
**Fontaine classification**			
No peripheral arterial disease, n (%)	98 (47%)		
Asymptomatic, n (%)	67 (32%)		
Symptomatic, n (%)	42 (20%)		
**Ophthalmology**	***n* = 193**		
Length of angioid streaks			
<3 mm from optic disc	1 (0.5%)		
3–6 mm from optic disc	38 (20%)		
6–9 mm from optic disc	60 (32%)		
>9 mm from optic disc	87 (47%)		
Choroidal neovascularization, n (%)	119 (62%)		
Macular atrophy, n (%)			
Mild	25 (13%)		
Severe	29 (15%)		
Visual acuity, LogMar	0.10 (0.00–0.70)		

Data are shown as mean ± SD, median [IQR] or n (%). Differences between the PXE patients and controls were analyzed with Student’s *t*-test (age, PP_i_) or the χ^2^ test (sex). A *p*-value < 0.05 was regarded as statistically significant. The length of angioid streaks was missing in seven PXE patients. In two patients, 55 near-infrared reflectance imaging was missing, while in five patients severe scarring or atrophy impaired reliable grading.

**Table 2 jcm-12-01047-t002:** Association between plasma PP_i_ and different arterial and ophthalmological phenotypes.

Variable	Age and Sex Adjusted β (95% Confidence Interval)
**Arterial calcification mass**	
Peripheral arteries	0.00 (−0.02;0.02)
Total body	0.01 (−0.02;0.04)
**Arterial functioning**	
Ankle brachial index	0.02 (−0.06;0.09)
Peripheral arterial disease	0.02 (−0.02;0.06)
Fontaine classification	0.00 (−0.03;0.03)
**Ophthalmology**	
Angioid streaks	0.01 (−0.02;0.03)
Choroidal neovascularization	0.06 (0.01;0.11) *
Macular atrophy	0.01 (−0.02;0.04)
Visual acuity	−0.00 (−0.04;0.03)

Linear regression models were built with one of the above-mentioned variables as the determinant and plasma PP_i_ in µM as the outcome. Models were adjusted for age and sex. Genetics = two truncating, mixed and two non-truncating variants. * = *p*-value < 0.05.

## Data Availability

The data presented in this study are available in Appendix A.

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
