# Peer review of "Plasma Level of Pyrophosphate Is Low in Pseudoxanthoma Elasticum Owing to Mutations in the ABCC6 Gene, but It Does Not Correlate with ABCC6 Genotype"

_jcm, 2023, doi:10.3390/jcm12031047_

Round 1
Reviewer 1 Report
The manuscript investigates the level of PPi in PXE patients and correlates results with the cardiovascular and ophthalmological phenotype. PPi levels do not correlate with disease severity and ABCC6 genotype.
The lack of correlation between PPi levels and PXE cardiovascular manifestations has been already demonstrated, in the present study data are confirmed and supported by additional data.
Authors should consider the following points:
1 )Line 159: “CNV and macular atrophy were scored in both eyes”. However, in line 188 “... data of the right eye was used for descriptive and regression analyses. If imaging of the right eye was not assessable, the left eye was used”. It is not clear how data were collected and analyzed. Authors should specify. Moreover, clinical manifestations are frequently different depending on the eye, therefore I would be important to correlate PPi levels with mean values obtained from both eyes.
2) Line 207: Authors state that PPi level was moderately correlated with age. Data are significant or not significant, It is not clear the meaning of “moderately”. Furthermore, disease severity correlates with aging, but not with the severity of the clinical phenotype. Please, explain better.
3) Figure 4 pane B: Figure is clear. In particular I do not understand how changes in the first box are related with no correlation between genotype ABCC6 mutation and plasma PPi. Why ABCC6, ENPP1 and ANK are reported as increased? Moreover, the term ABCC6 activity refers to ABCC6 gene expression or to protein activity? If Authors are referring to protein activity, they should more correctly indicate MRP6 activity.
4) Lines 290-399: It is not clear the meaning of the last part of the discussion. It is the list of possible therapeutic treatment, without any direct explained relationship with results provided in the manuscript. On the contrary, it would be more interesting to refer to previous studies indicating that low levels of PPi are not the exclusive cause of calcification. See for instance: 1) J Invest Dermatol. 2017;137:2336-2343, 2) J Invest Dermatol. 2019;139:360-368.
5) Check that all reference are appropriately attributed to sentence/paragraph. For instance reference 22 (J Biol Chem 2010;285:22800-8) does not seem to be the most appropriate to explain the effect of oxidative stress. There are several papers more suitable to support the concept. I would recommend Authors not to exceed self-references.
6) Revision of the English language is required
Reviewer 2 Report
1-English language and style are fine/minor spell check required
2-Introduction does not provide sufficient background and not include all relevant references
3-References need to be renumbered within the manuscript
4-A large number of references belong to the same researchers for example 2,5,7,10, 4, 17,18,20,22,29,30
Round 2
Reviewer 1 Report
The manuscript has been revised in some parts based on specific reviewer’s suggestions, however they did not revise accordingly the whole text.
I will try to be more detailed:
1) ABCC6 is the name of the gene, whereas the encoded protein, according also to all databases, is MRP6. Therefore, it is possible to measure the activity of the protein (MRP6) and the expression of the gene (ABCC6). Authors actually did NOT measure protein activity nor gene expression. In the text, ABCC6 is used indifferently as a gene and a protein name. ABCC6 is used to identify the protein in the following lines: 57,58,290, 295, 299, 304, 325, 347, 349, 353, 362, 376, 378. It is not sufficient to make changes in line 353 and not in the whole text.
If Authors do not want to make the appropriate distinction between gene and protein, since in most cases the name of the protein is the same of the gene, at least they should make a clear statement that, in order to oversimplify, ABCC6 in the manuscript is attributed to both the gene and to the encoded protein.
2) Authors are probably not familiar with the correct nomenclature, but SNPs are, as the name says, polymorphisms (frequency > 1%) and not rare pathogenic variants (frequency <1%). Reading line 85, it seems that Sanger sequencing was done to reveal only SNPs ( i.e. NOT mutations). Authors should add “rare sequence variants” before SNPs if they were also interested in polymorphisms, or they should replace “single nucleotide polymorphisms (SNPs)” with rare sequence variants.
3) Line 400: Authors state that INZ-701 increased PPi levels and prevented calcification in the Abcc6-/- mouse model (ref 33). It would be more correct to state “partially prevented calcification”. Actually, calcification, although reduced, was still present and, as already underlined, PPi is not the only cause of calcification.
Moreover, in the light of these data, Authors should comment more deeply and critically the observation that high PPi levels (i.e. high levels of the inhibitor of calcification) are, for instance, associated with old age (more severe phenotype and calcification), in females (more frequently affected by calcification), with severe macular atrophy (more severe phenotype possibly as a consequence at older age of more severe calcification). Additional perturbing factors, as those in figure 4 panel b, simply represent a hypothesis, not based/sustained on results (enzyme/protein activities were never measured or investigated in the study), therefore it would be more appropriate and correct, if Authors remove the grey boxes in the figure, leaving what is derived from their study.
It would be of interest of the Authors would be able to prove it with some data.
